# Effects of *MSTN* Gene Knockout on Growth Performance and Muscle Transcriptome in Chinese Merino Sheep (Xinjiang Type)

**DOI:** 10.3390/ani15233387

**Published:** 2025-11-24

**Authors:** Li Zhang, Pengfei Li, Xu Wang, Menghua Kong, Weiwei Wu, Wenxin Zheng

**Affiliations:** 1College of Animal Science, Xinjiang Agricultural University, Urumqi 830091, China; 15293662719@163.com (L.Z.); pflty173@126.com (P.L.); 13274713940@163.com (X.W.); 15739393146@163.com (M.K.); 2Institute of Animal Husbandry, Xinjiang Academy of Animal Science, Urumqi 830009, China

**Keywords:** transcriptome, Chinese Merino sheep (Xinjiang type), myostatin (*MSTN*)

## Abstract

Myostatin (*MSTN*) is a key regulator of muscle development in animals, and its knockout has been shown to significantly enhance growth performance. In this study, *MSTN* knockout Chinese Merino sheep (Xinjiang type) were used to assess the effects of *MSTN* deletion on growth performance and the underlying molecular mechanisms. The results demonstrated that *MSTN* knockout significantly increased body weight, body length, and height, with no adverse effects on hematological or metabolic functions. Transcriptomic analysis identified 121 differentially expressed genes (DEGs), which were involved in critical pathways such as amino acid metabolism, muscle contraction, and immune response. A protein–protein interaction (PPI) network was constructed, revealing 10 key genes that may play pivotal roles in *MSTN* gene regulation. This study provides valuable theoretical support for gene editing in meat sheep breeding, with substantial practical implications.

## 1. Introduction

Myostatin, also known as Growth Differentiation Factor 8 (*GDF8*), is a member of the transforming growth factor-beta (TGF-β) superfamily and functions as a negative regulator of skeletal muscle development, playing a critical role in the growth and development of animals [1]. *MSTN* regulates muscle development by inhibiting the proliferation and differentiation of skeletal muscle satellite cells. Its deletion or functional defects result in significant muscle hypertrophy, leading to a characteristic “Double-Muscling” phenotype [2]. Studies have shown that *MSTN* is highly conserved across various livestock species, including cattle [3], sheep [4], pigs [5], and rabbits [6]. Mutations or knockout of *MSTN* have been shown to significantly improve meat production traits, emphasizing its potential in livestock genetic improvement and the enhancement of production performance.

With the advancement of molecular biology and genetic engineering, next-generation gene editing tools have enabled the targeted improvement of large animals. Among these tools, CRISPR/Cas-based systems have become key for livestock genetic enhancement due to their efficiency, precision, and flexibility [7]. In recent years, researchers have successfully achieved effective *MSTN* gene knockout in economically important animals such as pigs and sheep using CRISPR/Cas9, Cpf1, and Prime Editing technologies. These edits have resulted in increased muscle deposition and accelerated growth in the edited individuals [8,9,10]. These findings not only validate the feasibility of *MSTN*-directed genetic improvement for enhancing meat production traits but also offer new perspectives for molecular breeding in livestock. However, existing studies primarily focus on phenotypic growth traits [11], with limited systematic investigation at the hematological and transcriptomic levels. In particular, research on the molecular mechanisms underlying *MSTN*-edited sheep, a dual-purpose species for wool and meat, remains insufficient [12]. This gap has hindered the comprehensive application and wider adoption of these findings.

The Chinese Merino sheep (Xinjiang type) is a key breed in the northern pastoral regions of China, renowned for its high-quality fine wool and meat production potential. However, its growth rate and meat production performance are relatively limited [13]. Enhancing skeletal muscle deposition and increasing muscle mass through *MSTN*-directed breeding offers promising opportunities for the genetic improvement of this breed, providing both theoretical and practical value [14]. Building on this premise, this study utilized an optimized hfCas12Max-T5_sheep gene editing system, which has demonstrated high editing efficiency and accuracy in both sheep cells and individuals [15]. By employing *MSTN* knockout Chinese Merino sheep, this study systematically compared differences in growth performance, hematological indicators, and muscle transcriptome between the edited and control groups. The objective was to investigate the impact of *MSTN* knockout on sheep growth, development, and molecular regulatory mechanisms, thereby providing theoretical support and practical insights for gene editing in meat sheep breeding [16].

## 2. Materials and Methods

### 2.1. Experimental Animals and Housing Conditions

The animals used in this experiment were sourced from the Qitai Sheep Breeding Farm in Xinjiang. A total of 100 Chinese Merino sheep (Xinjiang type) were selected, comprising 50 *MSTN* gene-edited sheep (MT) and 50 wild-type control sheep (WT). The sheep were matched for sex and age, and all individuals were in good health. The gene-edited sheep were generated using T7-hfCas12Max-T5_sheep plasmid construction and embryo microinjection techniques. Successful knockout of the target site was confirmed by PCR, Sanger sequencing, and deep-targeted sequencing. The genotype distribution was as follows: among the 50 individuals in the MT group, 32 were homozygous mutants (64%) and 18 were heterozygous mutants (36%). No chimeric individuals were detected, and the mutations consisted of base substitutions or deletions.

All 100 Chinese Merino sheep (Xinjiang type) selected for this experiment were of the same generation and half-siblings, sharing the same sire but different dams, ensuring a consistent genetic background. The male-to-female ratio was 1:1 in both groups, with 25 rams and 25 ewes in each group. All individuals were intact (non-castrated). The ewes used for reproduction were primiparous, and all lambs were single-born, ensuring a consistent early growth environment for both groups and minimizing potential confounding effects.

The sheep were all raised on the same farm from birth under identical management conditions. During the lactation period, the lambs were exclusively fed by their mothers. From 28 days of age, supplementary feeding commenced, and after weaning, the lambs were fed a uniform formula total mixed ration (TMR) with unrestricted access to water. Throughout the experimental period, both groups of sheep were raised in the same environment with identical diets and management practices to ensure the scientific validity and comparability of the results. All sampling and handling procedures adhered strictly to ethical guidelines for animal experiments, ensuring the welfare of the animals.

### 2.2. Growth Performance Measurement

Growth performance was assessed at eight time points: birth, 30 days, 60 days, 90 days, 120 days, 160 days, 190 days, and 365 days. The traits measured included body weight (kg), withers height (cm), body length (cm), chest girth (cm), cannon circumference (cm), and hip width (cm). Body weight was recorded using an electronic scale to ensure precise measurements, while body dimensions were taken by a single trained operator to minimize inter-observer variability. All measurements were performed in the morning, prior to feeding, to ensure consistency and accuracy across time points.

### 2.3. Blood Sample Collection and Analysis

Blood samples were collected in both heparinized and non-heparinized tubes. The non-heparinized blood was allowed to stand at room temperature to facilitate serum separation, then centrifuged at 3000 rpm for 10 min at 4 °C. The serum was carefully collected and transferred to a 2 mL sterile centrifuge tube. Both serum and heparinized whole blood samples were then sent to Zhongchu Animal Hospital in Urumqi, Xinjiang, for analysis. Hematological analysis was conducted using an automatic veterinary hematology analyzer (AL-90Vet, High Technology Inc., Beijing, China), which provided 46 hematological parameters. Serum biochemical analysis was carried out using an automatic dry chemistry analyzer (Seamaty Minilab VET, Chengdu Seamaty Technology Co., Ltd., Chengdu, China), yielding 24 serum biochemical parameters.

### 2.4. Muscle Tissue RNA Extraction, Transcriptome Library Construction, and Sequencing

Muscle tissue from the same region of the hind limbs was collected from 3 *MSTN*-edited Chinese Merino sheep (Xinjiang type) and 3 wild-type Chinese Merino sheep (Xinjiang type) using in vivo sampling techniques. The samples were immediately stored in liquid nitrogen. Total RNA was extracted using the TRIzol method (Thermo Fisher Scientific, Waltham, MA, USA), and RNA concentration and purity were measured with a NanoDrop 2000 spectrophotometer (Thermo Scientific, Waltham, MA, USA). RNA integrity was assessed using an Agilent 2100 Bioanalyzer (Agilent Technologies, Santa Clara, CA, USA).

Transcriptome libraries were constructed with the VAHTS Universal V10 RNA-seq Library Prep Kit (Premixed Version) following the manufacturer’s instructions. Library sequencing was performed by Shanghai OE Biotech Co., Ltd. (Shanghai, China) using the Illumina NovaSeq 6000 sequencing platform with 150 bp paired-end sequencing. Following quality control using fastp software (v0.23.4), the sequencing data were aligned to the sheep reference genome (ARS-UI_Ramb_v2.0) with HISAT2 software (v2.2.1), and gene expression levels (FPKM) were calculated. Read counts were obtained using HTSeq-count (v2.0.5).

### 2.5. Identification of Differentially Expressed Genes and Functional Enrichment Analysis

Principal component analysis (PCA) and plotting were performed using R (v3.2.0) to evaluate the biological reproducibility of the samples. Differentially expressed genes (DEGs) were identified using DESeq2 (v1.40.2), with thresholds set to padj < 0.05 and |log2FoldChange| > 1. Gene Ontology (GO) and Kyoto Encyclopedia of Genes and Genomes (KEGG) enrichment analyses were conducted on the DEGs. Volcano plots were generated using the R package ggplot2 to visualize the upregulated and downregulated gene expression changes. Additionally, enrichment circle plots were created to display significantly enriched functional terms.

### 2.6. Protein–Protein Interaction Network Construction

To investigate potential interactions between proteins encoded by the differentially expressed genes (DEGs), a protein–protein interaction (PPI) network was constructed using DEGs with padj < 0.05 and |log2FoldChange| ≥ 1. The corresponding genes of the DEGs were first converted to UniProt or Ensembl protein IDs, and protein interaction data were retrieved from the STRING database (v11.5, https://string-db.org, accessed on 6 August 2025), retaining only high-confidence interactions (confidence score ≥ 0.7). The interaction data were then imported into Cytoscape (v3.9.1) to construct the PPI network. Network topology metrics were calculated using the Network Analyzer tool to identify hub proteins. Core modules within the network were further identified using the MCODE plugin. GO and KEGG enrichment analyses were conducted on the proteins within these modules to elucidate the potential roles of the DEGs in key biological processes. This approach visually presents the interaction relationships and functional characteristics of DEG-encoded proteins, offering valuable insights for subsequent mechanistic studies.

### 2.7. Real-Time Quantitative PCR (RT-qPCR) Validation

To validate the RNA-Seq differential expression results, total RNA was extracted from the same batch of muscle samples using the TRIzol method (Thermo Fisher Scientific, Waltham, MA, USA), and RNA purity and concentration were assessed using the NanoDrop 2000. Genomic DNA was removed, and cDNA was synthesized using the PrimeScript™ RT Reagent Kit (Takara Bio Inc., Kusatsu, Shiga, Japan). RT-qPCR was performed using the SYBR Green method on the QuantStudio 6 Flex real-time PCR system (Thermo Fisher Scientific, Waltham, MA, USA). Primers for target and housekeeping genes were designed to span exon-exon junctions to ensure specificity. Each sample was measured in triplicate, and reactions were carried out according to the manufacturer’s recommended conditions. Relative gene expression levels were calculated using the 2^−ΔΔCT^ method, with significance determined at *p* < 0.05.

## 3. Results

### 3.1. Effects of MSTN Knockout on Body Weight and Body Measurements

To systematically assess the impact of *MSTN* gene editing on the growth and development of sheep, this study conducted dynamic monitoring of body weight and body measurements from birth to 365 days of age in both MT and WT groups. At birth, the MT group had a significantly higher weight of 3.98 ± 1.00 kg compared to the WT group, which weighed 3.47 ± 0.77 kg (*p* < 0.05). At 30, 120, and 160 days of age, the MT group exhibited significantly greater body weight, body length, body height, chest girth, cannon circumference, and hip width (*p* < 0.05), indicating faster growth and skeletal development. At 60 days of age, the MT group showed significantly higher body weight, body length, body height, and hip width compared to the WT group (*p* < 0.05), while no significant differences were observed in chest girth and cannon circumference (*p* > 0.05). By 90, 190, and 365 days of age, the MT group continued to show significantly higher body weight, body length, body height, chest girth, and hip width compared to the WT group (*p* < 0.05), with no significant difference in cannon circumference (*p* > 0.05) (Table 1). These results demonstrate that *MSTN* gene knockout significantly promotes early growth and development in sheep, particularly in terms of muscle mass accumulation and skeletal traits, highlighting its important application in accelerating meat sheep breeding and enhancing production performance.

### 3.2. Effects of MSTN Gene Editing on Hematological and Serum Biochemical Parameters in Sheep

To assess the potential impact of *MSTN* gene editing on the health status of sheep, this study analyzed the hematological and serum biochemical parameters in both the experimental (MT) and control (WT) groups. There were no significant differences between the MT and WT groups in terms of hematological parameters, including white blood cell count, absolute neutrophil count, absolute lymphocyte count, red blood cell count, hemoglobin concentration, hematocrit, mean corpuscular volume, mean corpuscular hemoglobin concentration, platelet count, and mean platelet volume (*p* > 0.05) (Table 2). Other hematological indicators are presented in Appendix A. Regarding serum biochemical parameters, no significant differences were observed between the experimental and control groups in albumin, total protein, albumin/globulin ratio, aspartate aminotransferase, alanine aminotransferase, lactate dehydrogenase, creatine kinase, creatinine, urea, glucose, total cholesterol, triglycerides, and calcium concentration (*p* > 0.05) (Table 3). Other serum biochemical indicators are shown in Appendix A. In conclusion, *MSTN* gene-edited sheep exhibited normal hematological and serum biochemical parameters with no significant differences compared to the control group, indicating that gene editing did not cause any notable adverse effects on their health or metabolic functions.

### 3.3. Statistical Analysis of Sequencing Data

RNA-Seq was performed on six samples, generating approximately 284 million raw paired-end 150 bp reads, with an average of 47.1 million reads per sample. A total of 41.44 GB of clean data was obtained, with the effective data volume for each sample ranging from 6.78 to 7.06 GB. After quality control using fastp software, each sample retained approximately 46.47 million high-quality clean reads, with the proportion of valid bases exceeding 96.99%. The Q30 base percentage of clean reads ranged from 96.69% to 96.97%, indicating high sequencing data quality. The GC content of the samples ranged from 55.83% to 56.68%, which is consistent with the expected GC distribution of the sheep genome. Subsequent alignment of the clean reads to the sheep reference genome (ARS-UI_Ramb_v2.0) resulted in a total alignment rate of 97.83% to 99.06%, with correctly paired reads accounting for 91.80% to 92.87%, reflecting good alignment efficiency and suitability for the reference genome (Table 4). Additional data quality control results are presented in Appendix A. In conclusion, the RNA-Seq data obtained are of reliable quality and meet the requirements for subsequent differential expression analysis and functional annotation.

### 3.4. Sample Correlation Analysis

To evaluate the overall consistency and reproducibility between sequencing samples, Pearson correlation coefficients were calculated for the normalized gene expression matrix (FPKM) of each sample (Appendix A), and a correlation heatmap was generated. The correlation coefficients (R^2^) between samples within the same group were all above 0.86, significantly higher than those between different groups (Figure 1A), indicating good internal consistency among biological replicates and reliable data quality, which meets the requirements for subsequent differential expression analysis. Furthermore, principal component analysis (PCA) showed clear clustering separation of samples from different experimental groups along the first and second principal components (Figure 1B), further supporting the existence of inter-group differences and the stability of intra-group replicates.

### 3.5. Differential Gene Screening

A total of 121 differentially expressed genes (DEGs) were identified, with 54 genes showing upregulation and 67 genes showing downregulation (|log_2_ Foldchange| ≥ 1, padj < 0.05) (Figure 2, Appendix A). The distribution of these DEGs was relatively uniform, with distinct trends of upregulation and downregulation, indicating significant transcriptional differences between the MT and WT groups in muscle tissue. These findings suggest that the identified DEGs may be closely linked to biological processes such as muscle growth regulation, metabolic pathways, and cell signaling mechanisms induced by *MSTN* gene knockout. This provides a solid data foundation for subsequent Gene Ontology (GO) and Kyoto Encyclopedia of Genes and Genomes (KEGG) functional enrichment analyses.

### 3.6. GO and KEGG Functional Enrichment of Differentially Expressed Genes

Based on the identified differentially expressed genes (DEGs), Gene Ontology (GO) and Kyoto Encyclopedia of Genes and Genomes (KEGG) functional enrichment analyses were conducted to further explore their potential biological functions and associated signaling pathways. The GO enrichment results revealed that, at the biological process (BP) level, the DEGs were predominantly enriched in the negative regulation of oxidative phosphorylation (GO:0090324), RNA polymerase II-mediated transcription (GO:0006357), the urea cycle (GO:0000050), calcium ion transmembrane transport (GO:0070588), and ischemic response (GO:0002931). At the cellular component (CC) level, the DEGs were mainly localized to the sarcolemma (GO:0042383). At the molecular function (MF) level, DEGs were significantly involved in oxidoreductase activity (GO:0016491) (Figure 3A, Detailed FDR values and gene counts refer to Appendix A). These results suggest that *MSTN* knockout may affect skeletal muscle development and physiological function by regulating energy metabolism, ion homeostasis, and stress responses.

KEGG pathway analysis further demonstrated that the DEGs were significantly enriched in metabolic pathways related to arginine biosynthesis (oas00220), amino acid biosynthesis (oas01230), and cysteine and methionine metabolism (oas00270). Additionally, DEGs were involved in pathways related to reactive oxygen species (ROS)-mediated chemical carcinogenesis (oas05208), muscle cell cytoskeleton structure (oas04820), and overall metabolic pathways (oas01100) (Figure 3B, Detailed FDR values and gene counts refer to Appendix A). These findings indicate that *MSTN* knockout may not only provide the energy and material basis for muscle growth through amino acid metabolism and redox reactions but also regulate skeletal muscle structure and function through cytoskeletal remodeling and stress response. This provides a theoretical basis for further elucidating the molecular mechanisms and identifying key targets for future studies.

### 3.7. Protein–Protein Interaction Network Construction and Analysis

Based on the 121 differentially expressed genes (DEGs) identified in this study, a protein–protein interaction (PPI) network was constructed using the STRING database, and visualized and modularized in Cytoscape. The initial network contained 107 nodes (including isolated nodes). To improve the clarity of the network, 63 isolated nodes that did not interact with other proteins were removed, resulting in a high-confidence interaction network consisting of 44 nodes and 57 interaction edges. Network topology analysis revealed that the PPI network could be divided into three relatively independent functional modules (Figure 4A–C): Module 1 was primarily enriched in amino acid metabolism and mitochondrial-related proteins; Module 2 focused on muscle contraction, excitation-contraction coupling, and ion channel-related proteins; Module 3 was enriched in interferon-mediated antiviral response-related proteins.

Furthermore, using the Cytoscape plugin cytoHubba and sorting nodes by their Degree values, the top 10 hub genes with the highest connectivity in the network were identified, including *IRF7* (Interferon Regulatory Factor 7), *ISG15* (Interferon-Stimulated Gene 15), *CACNA1S* (Calcium Voltage-Gated Channel Subunit Alpha1 S), *MX1* (MX Dynamin-Like GTPase 1), RSAD2 (Radical S-Adenosyl Methionine Domain Containing 2), *GOT1* (Glutamic-Oxaloacetic Transaminase 1), *CLCN1* (Chloride Voltage-Gated Channel 1), *RYR1* (Ryanodine Receptor 1), *EIF2AK2* (Eukaryotic Translation Initiation Factor 2 Alpha Kinase 2), and *MX2* (MX Dynamin-Like GTPase 2). These genes are located at the core of the network and are likely to play crucial roles in the biological processes regulated by *MSTN*, providing potential candidate targets for further molecular mechanism exploration and functional validation.

### 3.8. RT-qPCR Validation

To systematically verify the reliability and accuracy of the transcriptome sequencing data, four key candidate genes (*ACTN3*, *RYR1*, *MX1*, *IRF7*) were randomly selected from the screened differentially expressed genes (DEGs) for validation by quantitative real-time PCR (RT-qPCR) (Figure 5), with primer sequences provided in Appendix A. Glyceraldehyde-3-phosphate dehydrogenase (*GAPDH*) was used as the reference gene to eliminate systematic variations in RNA extraction efficiency, reverse transcription efficiency, and PCR amplification efficiency among samples. Each candidate gene was set with 3 biological replicates (exactly consistent with the transcriptome sequencing samples: 3 *MSTN* knockout sheep and 3 wild-type sheep), and each biological replicate was subjected to 3 technical replicates to ensure the reproducibility and robustness of the experimental results. Determined by the standard curve method, the PCR amplification efficiencies of all candidate genes and the reference gene ranged from 95% to 105%, an ideal range, as follows: *ACTN3* (97.3%), RYR1 (98.6%), *MX1* (101.2%), *IRF7* (96.8%), and *GAPDH* (100.5%). Melting curve analysis showed that all primers exhibited a single sharp peak without non-specific peaks or primer dimers, confirming good primer specificity. The relative expression levels of the candidate genes were calculated using the 2^−ΔΔCt^ method. The results indicated that the expression trends and relative fold changes of the four candidate genes between the *MSTN* knockout group and the wild-type group were consistent with the transcriptome sequencing data. These findings fully confirm the reliability, accuracy, and biological authenticity of the transcriptome sequencing data in this study.

## 4. Discussion

*MSTN* knockout significantly enhanced the growth performance of Chinese Merino sheep (Xinjiang type). Body weight and body measurements (e.g., body length, body height, chest circumference, and rump width) were significantly superior to those of the control group across multiple growth stages. This was particularly evident for traits that directly reflect skeletal muscle deposition and individual development, such as body length, body height, chest circumference, and rump width [17,18]. These advantages persisted from birth through 365 days of age. Consistent with findings from *MSTN* editing studies in cattle, pigs, goats, and other livestock, these results confirm the cross-species conservation of *MSTN* as a negative regulator of skeletal muscle growth [19,20]. The core mechanism involves the loss of *MSTN*, which alleviates the inhibition of skeletal muscle satellite cell proliferation and differentiation, resulting in increased muscle fiber number and diameter. This may also synergistically promote bone growth, ultimately contributing to the simultaneous improvement of body weight and body measurements [21]. This sustained growth advantage indicates that *MSTN* knockout not only optimizes early muscle deposition but also supports continued performance enhancement throughout the entire growth cycle, providing a solid theoretical foundation for long-term improvement in meat sheep breeding.

Carcass traits are key evaluation indicators for the production efficiency of meat sheep. The positive effects of *MSTN* knockout on carcass performance have been widely validated across various livestock species, providing valuable references for the application prospects of the breed in this study. In studies on goats and sheep, *MSTN* deficiency significantly increased dressing percentage and net meat percentage, with a 15–25% increase in longissimus dorsi muscle area [22]. These improvements are strongly correlated with the significant gains in body length, chest circumference, and other body measurements observed in this study. Given the sustained muscle growth advantage of *MSTN*-edited sheep, it is hypothesized that their lean meat production efficiency post-slaughter will be significantly enhanced, and key carcass indicators, such as longissimus dorsi muscle area, are expected to show similar improvements.

Regarding fat deposition, *MSTN* knockout has been shown to preferentially direct nutrient allocation to skeletal muscle, reducing the proportion of subcutaneous and visceral fat [23]. For example, *MSTN* knockout models in pigs exhibit high-quality carcass characteristics, including optimal intramuscular fat content and reduced backfat thickness [24]. This trait is particularly relevant for Chinese Merino sheep (Xinjiang type), a dual-purpose breed for both wool and meat. Moderately reducing fat deposition can improve lean meat percentage, aligning with market demands for high-quality lean meat. However, it is crucial to strike a balance in fat deposition to prevent potential adverse effects on meat quality due to excessive reduction in intramuscular fat content. The activation of amino acid metabolism pathways, as revealed in the transcriptomic analysis of this study, may help optimize nutrient allocation, maintaining a reasonable level of intramuscular fat while enhancing lean meat percentage. This provides potential for the coordinated improvement of carcass quality, contributing to the overall optimization of meat production traits.

Meat quality (including meat color, tenderness, flavor, and juiciness) is a central consideration in breeding practices, and the impact of *MSTN* knockout on meat quality exhibits species-specific variation. Existing studies have shown that *MSTN* knockout sheep display reduced shear force in the longissimus dorsi muscle, with no significant change in intramuscular fat content, while maintaining high-quality meat color. In this study, the observed improvement in tenderness may be linked to the differential expression of key genes in the calcium signaling pathway, such as *RYR1* and *CACNA1S*. As a sarcoplasmic reticulum calcium release channel gene, changes in *RYR1* expression can influence calcium ion recovery efficiency following muscle contraction, thereby improving muscle tenderness [25]. Additionally, the activation of amino acid metabolism pathways may provide precursors for the synthesis of flavor compounds, such as glutamic acid and inosinic acid [26], which could enhance meat flavor. However, potential risks should be carefully considered. In some species, *MSTN* knockout has been shown to cause excessively large muscle fiber diameters, which may negatively affect meat tenderness [27]. Moreover, if intramuscular fat content is too low, it could impair the juiciness and flavor balance of the meat [28]. Therefore, further verification is needed for this breed. Subsequent studies should systematically assess shear force, meat color (Lab* values), intramuscular fat content, fatty acid composition, and flavor metabolic profiles in *MSTN*-edited sheep. This comprehensive evaluation will help clarify the overall impact of gene editing on meat quality and provide robust support for breeding applications.

The blood routine and serum biochemical indicators of *MSTN*-edited sheep were within the normal range, showing no significant differences from the control group. Consistent with studies on pigs [29], these findings suggest that gene editing did not cause adverse effects on hematopoietic function, immune homeostasis, or liver and kidney metabolism, effectively achieving precise improvement of target traits [30]. Some indicators, such as creatine kinase and lactate dehydrogenase, were slightly elevated in the edited group, which may be related to enhanced muscle metabolism and increased energy consumption [31]. *MSTN* knockout may be associated with a mild increase in metabolic load while promoting muscle hypertrophy, a phenomenon confirmed in studies on double-gene-edited sheep [32]. Overall, the stability of hematological and biochemical indicators provides important support for the safety of *MSTN* gene editing technology.

At the molecular level, transcriptomic analysis revealed significant changes in 121 differentially expressed genes (DEGs), which were involved in several biological processes, including amino acid metabolism, muscle contraction, and immune response. Amino acid metabolism is a central process in skeletal muscle synthesis and energy supply. The upregulation of related genes suggests that *MSTN* deficiency may enhance amino acid uptake and utilization, thereby providing more substrates and energy support for muscle tissue [33]. Previous studies have shown that *MSTN*-deficient animals generally exhibit higher protein deposition efficiency and nitrogen deposition rate, consistent with the findings of this study [34]. Further analysis highlighted significant changes in the expression of calcium ion channel-related genes such as *RYR1*, *ATP2A1*, and *CACNB1* following *MSTN* knockout. As key signaling molecules for muscle contraction and relaxation, alterations in calcium ion homeostasis directly impact muscle fiber type transformation, contraction efficiency, and energy metabolism. Existing studies have pointed out that *RyR1* is crucial during myotube fusion and maturation, with its dysfunction hindering muscle differentiation [35,36]. *ATP2A1* (SERCA1) activity has been linked to pork quality traits, suggesting that its regulation of calcium recovery significantly impacts muscle relaxation and contraction rate [37]. Additionally, the expression and voltage sensitivity of the Ca_V1.1 channel complex, including its regulatory subunit *CACNB1*, play a central role in excitation-contraction coupling in skeletal muscle [38].

This study also identified significant changes in the expression of immune-related genes, such as *IRF7* and *ISG15*. It is noteworthy that skeletal muscle, in addition to being the primary organ for movement and metabolism, can function as an “immunoneuroendocrine organ,” secreting various myokines involved in inflammatory responses and immune signaling. As a core transcription factor in the type I interferon signaling pathway, the upregulation of *IRF7* suggests that *MSTN* deficiency may activate antiviral or immune–regulatory pathways, thereby influencing immune–metabolic coupling [39]. Similarly, *ISG15*, a typical interferon-stimulated gene, plays a vital role in cellular stress responses and antiviral immunity. It regulates protein homeostasis through ISGylation and participates in muscle cell differentiation and maturation [40,41]. These results suggest that *MSTN* knockout not only alters muscle metabolic pathways but may also affect muscle homeostasis and adaptability by reconstructing immune networks. Previous studies have shown that *MSTN* cross-regulates inflammation-related pathways, such as NF-κB and interferons [42,43], and the results of this study provide new evidence supporting this hypothesis.

The results of this study are consistent with existing findings in cattle, pigs, and other animals. Following *MSTN* knockout, differentially expressed genes are primarily enriched in pathways related to amino acid metabolism and muscle energy metabolism, indicating that energy metabolism remodeling is a key mechanism promoting skeletal muscle hypertrophy. *MSTN* deficiency can not only directly relieve muscle growth inhibition but also indirectly promote myofiber hypertrophy by regulating energy metabolism and signal transduction [44]. Furthermore, *MSTN* has been shown to cross-regulate immune and inflammatory signals during muscle development, with muscle-secreted factors and calcium signaling pathways playing crucial roles in metabolic regulation and muscle structural remodeling [45]. The findings from this study in sheep suggest that *MSTN* knockout may synergistically promote rapid skeletal muscle growth and structural remodeling through the coordinated action of energy metabolism regulation, calcium signaling pathway reconstruction, and immune network modulation.

The promoting effect of *MSTN* knockout on the growth performance of various livestock species is highly conserved. It significantly increases body weight, skeletal muscle weight, and lean meat percentage in species such as cattle, sheep, pigs, and rabbits. At the transcriptomic level, *MSTN* knockout is enriched in amino acid metabolism, muscle contraction, and calcium signaling pathways [46,47], confirming that the molecular mechanism by which *MSTN* regulates muscle growth is highly conserved across mammals. These findings provide cross-species evidence supporting the reliability of the results in this study. However, species-specific differences are evident in the effects of *MSTN* editing on different livestock species. In terms of growth rate improvement, pigs and cattle show greater enhancements than sheep [48], which may be attributed to the genetic background and metabolic characteristics of sheep muscle growth. Regarding fat metabolism, the reduction in fat content is more pronounced in pigs [49], while the effect in sheep is milder [50], which may help maintain meat quality stability. At the molecular level, the immune-related core genes identified in this study, such as *IRF7* and *ISG15*, were not significantly enriched in studies on cattle and pigs, suggesting that *MSTN* may influence muscle growth through a unique immune–metabolic cross-regulation network in sheep. This species-specific characteristic warrants further in-depth exploration.

Although this study systematically confirmed the significant weight gain and body shape improvement associated with *MSTN* gene knockout in Chinese Merino sheep (Xinjiang type), without causing adverse effects on hematological and biochemical indicators, transcriptomic data further revealed that *MSTN* may regulate muscle growth through molecular mechanisms such as energy metabolism, calcium signaling pathways, and immune networks. These findings provide important theoretical support and practical references for the genetic improvement of this breed and for sheep molecular breeding practices. However, certain limitations remain:

First, the risk of off-target effects has not been completely excluded. While the efficient and precise hfCas12Max-T5_sheep system was used and no chimeric individuals were detected, the absence of whole-genome off-target site screening prevents a definitive conclusion regarding potential non-targeted mutations. Second, the study’s cycle and scope are limited. Currently, the focus is on growth and basic physiological indicators up to 365 days of age, lacking data on adult growth, reproductive indicators (such as estrous cycle, conception rate, and litter size), and comprehensive meat quality assessments. Third, no targeted animal welfare evaluation has been conducted. Excessive muscle hypertrophy may affect the sheep’s exercise capacity and bone load, which requires further verification.

To address these limitations, future studies will focus on the following aspects: First, conduct whole-genome off-target site screening to systematically evaluate the potential impact of non-target mutations, thereby ensuring the safety of gene editing technology. Second, extend the feeding cycle to monitor adult growth performance, reproductive performance, and the growth and development of offspring in edited sheep, and assess the genetic stability of the edited traits. Third, supplement comprehensive meat quality analysis, including shear force, meat color (Lab* values), intramuscular fat content, fatty acid composition, and flavor metabolic profiles, to clarify the overall impact of gene editing on meat quality. Fourth, systematically assess the actual impact of muscle hypertrophy on animal welfare by measuring gait parameters, cortisol levels, and bone mineral density. Fifth, conduct in-depth research on the effects of gene editing on grazing adaptability, cold resistance in winter, and forage conversion rate in the context of the pastoral breeding practices of Chinese Merino sheep (Xinjiang type), aiming to balance genetic improvement with ecological sustainability. Sixth, explore synergistic editing strategies combining *MSTN* with functional genes such as *FGF5* to optimize both wool and meat performance in this breed.

## 5. Conclusions

This study demonstrated that *MSTN* gene knockout significantly enhanced the growth performance of Chinese Merino sheep (Xinjiang type), with notable improvements in body weight, body shape, and muscle development. Gene editing did not cause any adverse effects on the hematological or metabolic functions of the sheep. Transcriptomic analysis identified 121 differentially expressed genes, primarily involved in amino acid metabolism and muscle contraction. Through protein–protein interaction (PPI) network analysis, 10 key genes were identified, which may play crucial roles in the molecular mechanisms underlying *MSTN* gene regulation. RT-qPCR validation further confirmed the reliability of these findings. This study provides important theoretical insights and practical references for the application of *MSTN* gene editing in meat sheep breeding.

## Figures and Tables

**Figure 1 animals-15-03387-f001:**
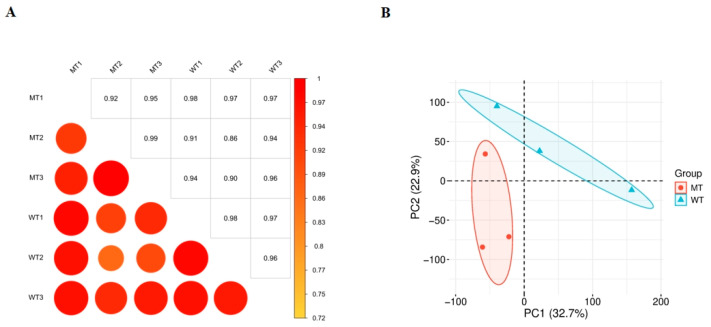
Pearson correlation analysis of samples (**A**) and principal component analysis of samples (**B**).

**Figure 2 animals-15-03387-f002:**
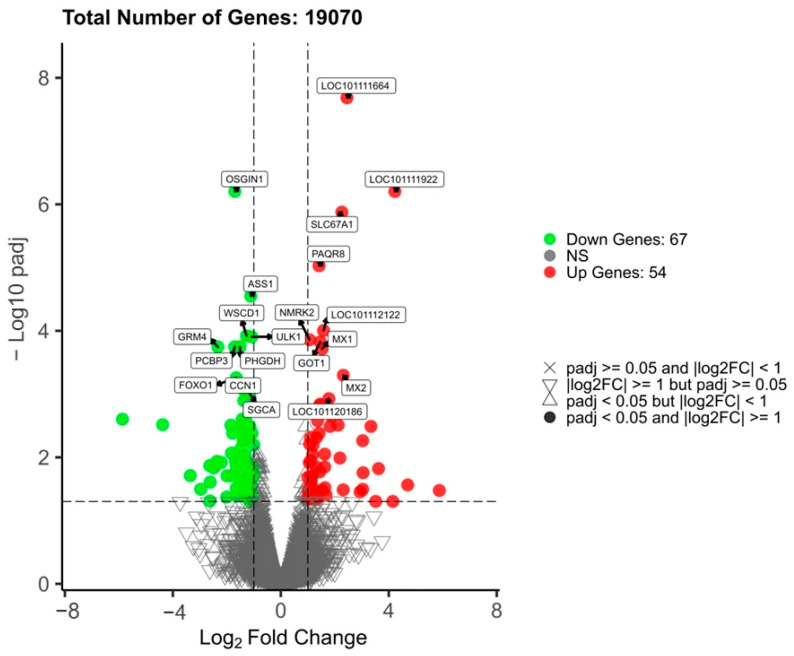
Volcano plot of differentially expressed genes.

**Figure 3 animals-15-03387-f003:**
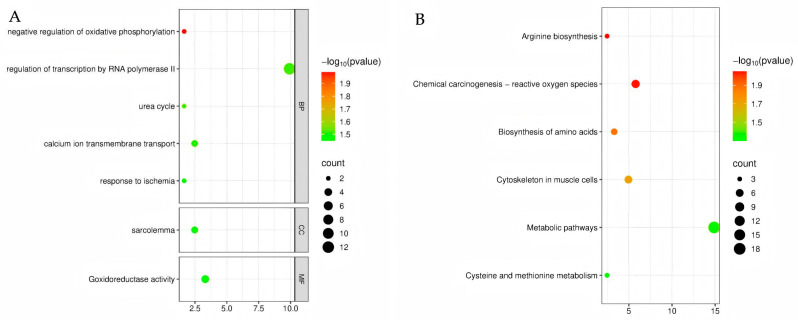
GO functional enrichment (**A**) and KEGG functional enrichment (**B**).

**Figure 4 animals-15-03387-f004:**
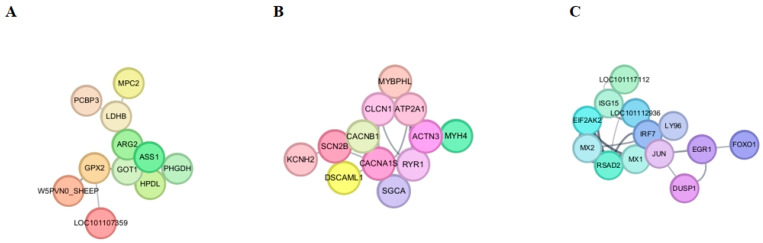
PPI network interaction analysis of differentially expressed genes (**A**–**C**). Panels (**A**–**C**) represent three distinct gene co-expression modules. Node colors indicate different functional clusters or gene families (specific color definitions are consistent with the legend in the original study).

**Figure 5 animals-15-03387-f005:**
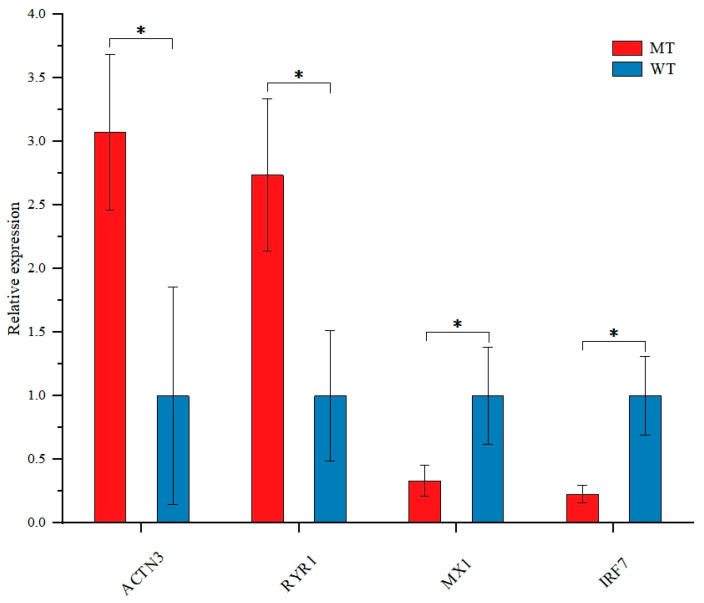
RT-qPCR validation of differentially expressed genes (DEGs). * *p* < 0.05 indicates a significant difference between groups.

**Table 1 animals-15-03387-t001:** Effects of *MSTN* gene knockout on body weight and body measurements at different stages of sheep development.

Age (Days)	Group	Body Weight (kg)	Body Length (cm)	Withers Height (cm)	Chest Girth (cm)	Cannon Circumference (cm)	Hip Width (cm)
30	MT	11.09 ± 3.16 a	43.23 ± 3.82 a	46.47 ± 3.60 a	51.14 ± 5.64 a	6.36 ± 0.60 a	12.82 ± 1.42 a
	WT	8.22 ± 1.48 b	37.79 ± 2.45 b	41.61 ± 2.45 b	42.44 ± 2.03 b	6.03 ± 0.38 b	10.62 ± 0.37 b
60	MT	18.32 ± 3.92 a	52.23 ± 4.46 a	53.17 ± 3.22 a	60.10 ± 5.40 a	6.63 ± 0.65 a	13.30 ± 1.64 a
	WT	16.07 ± 2.91 b	50.56 ± 3.64 b	50.82 ± 3.30 b	58.88 ± 4.52 a	6.43 ± 0.55 a	12.10 ± 1.16 b
90	MT	23.94 ± 4.52 a	59.38 ± 3.69 a	57.82 ± 3.70 a	67.82 ± 5.51 a	7.04 ± 0.56 a	17.31 ± 1.46 a
	WT	20.54 ± 2.92 b	56.88 ± 3.41 b	54.85 ± 2.24 b	65.28 ± 4.05 b	6.93 ± 0.52 a	15.17 ± 1.06 b
120	MT	28.52 ± 5.15 a	61.39 ± 4.62 a	58.81 ± 2.69 a	72.55 ± 5.18 a	7.05 ± 0.57 a	19.11 ± 1.58 a
	WT	24.11 ± 3.49 b	58.56 ± 3.71 b	56.38 ± 2.35 b	69.27 ± 4.60 b	6.78 ± 0.57 b	16.58 ± 1.21 b
160	MT	35.18 ± 5.85 a	66.22 ± 4.54 a	62.71 ± 3.14 a	78.46 ± 5.76 a	7.55 ± 0.73 a	21.74 ± 2.26 a
	WT	29.20 ± 3.71 b	60.24 ± 3.20 b	59.24 ± 3.36 b	74.62 ± 4.00 b	7.39 ± 0.59 a	18.46 ± 1.28 b
190	MT	38.39 ± 6.56 a	67.23 ± 4.35 a	64.00 ± 2.77 a	82.30 ± 5.40 a	7.70 ± 0.63 a	19.70 ± 1.63 a
	WT	31.93 ± 4.88 b	64.91 ± 4.11 b	61.58 ± 2.81 b	77.43 ± 4.35 b	7.52 ± 0.58 a	17.71 ± 1.28 b
365	MT	61.07 ± 10.59 a	80.85 ± 5.86 a	71.37 ± 4.48 a	113.16 ± 8.80 a	11.20 ± 1.35 a	32.40 ± 2.93 a
	WT	50.20 ± 7.97 b	75.60 ± 5.29 b	68.23 ± 4.06 b	104.70 ± 7.36 b	11.08 ± 1.14 a	27.50 ± 2.41 b

Note: All measurement data are presented as mean ± standard deviation (Mean ± SD). Differences in lamb growth traits (body weight, body length, withers height, chest girth, cannon circumference, and hip width) between groups at different ages were analyzed using one-way analysis of variance (ANOVA), followed by post hoc multiple comparisons. Different letters (ab) indicate significant differences between groups at the same age (*p* < 0.05), while the same letter indicates no significant difference between groups at the same age (*p* > 0.05).

**Table 2 animals-15-03387-t002:** Hematological parameters of gene-edited sheep (MT group) and control sheep (WT group).

Indicator	Unit	MT Group	WT Group	*p*-Value
White Blood Cell Count	109/L	7.35 ± 1.08	5.70 ± 1.18	0.752
Absolute Neutrophil Count	109/L	1.42 ± 0.35	1.32 ± 0.38	0.374
Absolute Lymphocyte Count	109/L	5.37 ± 0.94	4.02 ± 0.94	0.153
Red Blood Cell Count	1012/L	8.98 ± 0.75	8.30 ± 1.97	0.607
Hemoglobin Concentration	g/L	84.11 ± 10.49	85.71 ± 20.62	0.910
Hematocrit	%	27.83 ± 4.22	27.35 ± 7.41	0.926
Mean Corpuscular Volume	fL	30.91 ± 2.50	32.74 ± 1.33	0.325
Mean Corpuscular Hemoglobin Concentration	g/L	303.09 ± 9.70	315.61 ± 12.45	0.242
Platelet Count	109/L	168.60 ± 48.88	184.20 ± 75.61	0.779
Mean Platelet Volume	fL	6.48 ± 2.09	7.18 ± 4.26	0.813

**Table 3 animals-15-03387-t003:** Hematological and serum biochemical parameters of gene-edited sheep (MT group) and control sheep (WT group).

Indicator	Unit	MT Group	WT Group	*p*-Value
Albumin	g/L	36.56 ± 1.42	33.83 ± 2.28	0.153
Total Protein	g/L	72.40 ± 4.47	73.13 ± 2.64	0.819
Albumin/Globulin Ratio		1.02 ± 0.07	0.87 ± 0.18	0.240
Aspartate Aminotransferase	U/L	178.00 ± 38.31	139 ± 23.07	0.205
Alanine Aminotransferase	U/L	27.00 ± 3.46	25.33 ± 2.08	0.515
Lactate Dehydrogenase	U/L	595.00 ± 27.62	533.67 ± 62.05	0.193
Creatine Kinase	U/L	92.33 ± 11.24	73.33 ± 10.50	0.099
Creatinine	umol/L	118.07 ± 10.77	160.60 ± 46.97	0.201
Urea	mmol/L	5.78 ± 0.62	6.17 ± 0.51	0.455
Glucose	mmol/L	4.66 ± 0.47	4.97 ± 0.62	0.530
Total Cholesterol	mmol/L	1.56 ± 0.11	1.96 ± 0.36	0.134
Triglycerides	mmol/L	0.52 ± 0.23	0.58 ± 0.01	0.744
Calcium	mmol/L	2.40 ± 0.18	2.28 ± 0.08	0.378

Note: Differences in hematological and serum biochemical parameters between the two groups were analyzed using Student’s *t*-test, with results presented as *p*-values (*p* < 0.05 indicates a significant difference; *p* > 0.05 indicates no significant difference).

**Table 4 animals-15-03387-t004:** Sequencing data statistics.

Group	Sample	Total Reads	Total Mapped Reads	Clean Bases (G)	Valid Bases (%)	Q30 (%)	GC (%)
MT	1	46,957,018	46,440,752	6.87	97.54	96.90	56.68
	2	46,931,116	46,325,269	6.90	97.99	96.72	56.62
	3	47,106,932	46,083,379	6.93	97.98	96.69	56.05
WT	1	48,464,622	48,010,755	7.06	96.99	96.97	55.90
	2	47,057,204	46,444,644	6.90	97.72	96.79	55.83
	3	46,073,008	45,485,260	6.78	97.98	96.69	55.99

## Data Availability

The raw RNA-seq data have been deposited in the National Center for Biotechnology Information (NCBI) Sequence Read Archive (SRA) database, associated with Bio Project accession number PRJNA1367960 (https://www.ncbi.nlm.nih.gov/sra/, accessed on 23 November 2025). All other relevant raw data are available from the authors upon reasonable request.

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
