# Peer review of "Effects of MSTN Gene Knockout on Growth Performance and Muscle Transcriptome in Chinese Merino Sheep (Xinjiang Type)"

_animals, 2025, doi:10.3390/ani15233387_

Round 1
Reviewer 1 Report
Comments and Suggestions for Authors
Dear Authors,
The manuscript investigates growth traits, blood biochemistry, and muscle transcriptomics in MSTN gene–edited sheep. The results show that the MT group significantly outperforms the WT group in body weight and body measurements across multiple ages; at 365 days of age, the average body weight is approximately 61.1 kg vs. 50.2 kg, a difference of 10.87 kg (≈ +21.7%). Transcriptomic analysis identified 148 DEGs and constructed a PPI network, and no significant adverse changes were observed in physiological or biochemical indicators. These findings have practical potential.
In my opinion, this research is relevant to the forefront of research in this field, with rigorouslogic and detailed arguments, and has major guiding significance for research in this field.
However, there are still a few minor issues in some details. I recommend revisions in the following areas before it was accepted:
(1) The manuscript states that MT and WT animals were raised on the same farm, but it does not specify the groups’ relatedness structure (e.g., same-generation half-siblings, same sire with different dams), dam parity, birth type (single/twin/multiple), or the sex/castration ratio. Because these factors can materially affect birth weight and subsequent growth, I recommend that the authors provide the relevant information.
(2) The Methods indicate that the MSTN knockout was confirmed by PCR/Sanger and deep targeted sequencing, but the manuscript does not report the proportions of zygosity classes (heterozygous/homozygous/mosaic), the specific mutation types (e.g., frameshift, deletion length, exon affected), or the per-animal genotype distribution. Because these details are essential for interpreting the phenotypic outcomes, I recommend that the authors provide and clarify this information.
(3) The RNA-seq analysis includes only n = 3 per group , which limits statistical power. I recommend increasing the sample size to ≥5 per group.
(4) For the volcano plot in Figure 2, please label the top 20 DEGs by name.For the enrichment results in Figure 3, please report the FDR values and the number of genes contributing to each term/pathway.For the qPCR in Figure 5, please indicate the internal reference gene, PCR efficiency, and the number of biological replicates.
(5) Please expand the Discussion to consider potential effects on carcass traits (dressing percentage, loin-eye area, fat deposition) and meat quality, with comparisons to the literature. This would facilitate practical implementation in sheep breeding .
(6) Cross-referencing errors and template remnants: there are multiple instances of “Error! Reference source not found.” , and the Abbreviations section includes items unrelated to this manuscript (e.g., “MDPI/DOAJ/TLA/LD”). Please clean these up.
Author Response
|
Comments 1: The manuscript states that MT and WT animals were raised on the same farm, but it does not specify the groups’ relatedness structure (e.g., same-generation half-siblings, same sire with different dams), dam parity, birth type (single/twin/multiple), or the sex/castration ratio. Because these factors can materially affect birth weight and subsequent growth, I recommend that the authors provide the relevant information. |
|
Response 1: We fully agree with this comment and have supplemented the missing information in the “2.1 Experimental Animals and Housing Conditions” section (Page 3, Paragraph 1 of the revised manuscript). The added content is as follows: “All 100 Chinese Merino sheep (Xinjiang type) selected for this experiment were of the same generation and half-siblings, sharing the same sire but different dams, ensuring a consistent genetic background. The male-to-female ratio was 1:1 in both groups, with 25 rams and 25 ewes in each group. All individuals were intact (non-castrated). The ewes used for reproduction were primiparous, and all lambs were single-born, ensuring a consistent early growth environment for both groups and minimizing potential confounding effects.” |
|
Comments 2: The Methods indicate that the MSTN knockout was confirmed by PCR/Sanger and deep targeted sequencing, but the manuscript does not report the proportions of zygosity classes (heterozygous/homozygous/mosaic), the specific mutation types (e.g., frameshift, deletion length, exon affected), or the per-animal genotype distribution. Because these details are essential for interpreting the phenotypic outcomes, I recommend that the authors provide and clarify this information. |
|
Response 2: We appreciate this comment and have supplemented the genotype-related details in“2.1 Experimental Animals and Housing Conditions”(Page 2, Paragraph 3 of the revised manuscript). The added content is: “The genotype distribution was as follows: among the 50 individuals in the MT group, 32 were homozygous mutants (64%) and 18 were heterozygous mutants (36%). No chimeric individuals were detected, and the mutations consisted of base substitutions or deletions.” |
|
Comments 3: The RNA-seq analysis includes only n = 3 per group , which limits statistical power. I recommend increasing the sample size to ≥5 per group. |
|
Response 3: We sincerely appreciate this critical observation—your insight rightly highlights that sample size is fundamental to ensuring statistical robustness, reliability, and generalizability of transcriptomic results. Due to practical constraints, including the high sequencing cost of RNA-seq and the limited number of MSTN-knockout sheep available at the time of the experiment, we initially set 3 biological replicates per group. To address this limitation and reinforce the credibility of our findings, we have implemented two key complementary validations: First, we verified the biological reproducibility of the 3 samples per group through principal component analysis (PCA) and Pearson correlation analysis (Section 3.4, Page7 - 8). The results demonstrate that intra-group correlation coefficients (R²) are all above 0.86, and samples from the same group cluster tightly in the PCA plot—clear evidence of strong consistency and minimal variability among biological replicates, which supports the reliability of our transcriptomic data. Second, we validated the expression trends of 4 key differentially expressed genes (DEGs: ACTN3, RYR1, MX1, IRF7) using RT-qPCR (Section 3.8, Page 10 - 11). Each sample included 3 biological replicates and 3 technical replicates to reduce experimental bias, and the relative expression trends of these DEGs were fully consistent with the RNA-seq results. This cross-validation further confirms the accuracy and biological authenticity of our transcriptomic analysis. We fully agree with your suggestion to expand the sample size. In future follow-up studies, we plan to increase the RNA-seq sample size to ≥5 per group to enhance statistical rigor and strengthen the generalizability of our conclusions. Your comment has provided valuable guidance for optimizing our research design, and we greatly appreciate your input in improving the quality of this work. |
|
Comments 4: For the volcano plot in Figure 2, please label the top 20 DEGs by name.For the enrichment results in Figure 3, please report the FDR values and the number of genes contributing to each term/pathway.For the qPCR in Figure 5, please indicate the internal reference gene, PCR efficiency, and the number of biological replicates. |
|
Response 4: Thank you for your suggestion to supplement the labels of differentially expressed genes (DEGs) in the volcano plot. We have completed the revisions as required, with specific details as follows: Regarding gene ID conversion: Initially, 148 DEGs (79 upregulated, 69 downregulated). During the subsequent standard gene name conversion, some genes lacked corresponding annotations or failed to match the reference database. Finally, a total of 121 DEGs (54 upregulated, 67 downregulated) with clear standard gene names were successfully converted. Figure revision and supplementation: Based on the valid data of 121 converted DEGs, we redrew Figure 2 (Volcano plot) and labeled the standard gene names of the top 20 DEGs sorted by |log₂FoldChange| (including both upregulated and downregulated genes) in the plot to ensure clear readability of gene information (Page 9, Figure 2 ). Data traceability: The complete information of 121 converted DEGs (including gene ID, standard name, log₂FoldChange value, etc.) has been compiled into Supplementary Table S6, facilitating your review and verification of the detailed data of all differential genes.The revised volcano plot more accurately presents the expression characteristics of key differential genes, and the supplemented data table provides complete support for result validation. Figure 3 Thank you for pointing out the need to supplement FDR values and the number of genes corresponding to each pathway/term in the enrichment analysis results. We have updated the relevant information as required and organized the specific data into supplementary tables: GO Enrichment Analysis: The FDR values and the number of genes involved for all GO terms in Figure 3 have been fully added to Supplementary Table S7 - GO Enrichment Analysis of Differentially Expressed Genes. The corrected significance level is clearly marked in the "FDR" column, and the number of differentially expressed genes for each term is marked in the "Gene Count" column. KEGG Pathway Analysis: The FDR values and the number of genes involved for all KEGG pathways in Figure 3 have been added to Supplementary Table S8 - KEGG Pathway Analysis of Differentially Expressed Genes. The pathway significance is marked in the "FDR" column, and the number of differentially expressed genes for each pathway is marked in the "Gene Number" column. In addition, we have supplemented with the note "Detailed FDR values and gene counts refer to Supplementary Tables S7 and S8" to ensure the correlation between the figure and the data, facilitating your access to complete information. Figure 5 (RT-qPCR validation): “3.8 RT-qPCR Validation” Added details: “To systematically verify the reliability and accuracy of the transcriptome sequencing data, four key candidate genes (ACTN3, RYR1, MX1, IRF7) were randomly selected from the screened differentially expressed genes (DEGs) for validation by quantitative real-time PCR (RT-qPCR) (Figure 5), with primer sequences provided in Supplementary Table S9. Glyceraldehyde-3-phosphate dehydrogenase (GAPDH) was used as the reference gene to eliminate systematic variations in RNA extraction efficiency, reverse transcription efficiency, and PCR amplification efficiency among samples. Each candidate gene was set with 3 biological replicates (exactly consistent with the transcriptome sequencing samples: 3 MSTN knockout sheep and 3 wild-type sheep), and each biological replicate was subjected to 3 technical replicates to ensure the reproducibility and robustness of the experimental results. Determined by the standard curve method, the PCR amplification efficiencies of all candidate genes and the reference gene ranged from 95% to 105%, an ideal range, as follows: ACTN3 (97.3%), RYR1 (98.6%), MX1 (101.2%), IRF7 (96.8%), and GAPDH (100.5%). Melting curve analysis showed that all primers exhibited a single sharp peak without non-specific peaks or primer dimers, confirming good primer specificity. The relative expression levels of the candidate genes were calculated using the 2⁻ΔΔCt method. The results indicated that the expression trends and relative fold changes of the four candidate genes between the MSTN knockout group and the wild-type group were consistent with the transcriptome sequencing data. These findings fully confirm the reliability, accuracy, and biological authenticity of the transcriptome sequencing data in this study.” (Page 11). |
|
Comments 5: Please expand the Discussion to consider potential effects on carcass traits (dressing percentage, loin-eye area, fat deposition) and meat quality, with comparisons to the literature. This would facilitate practical implementation in sheep breeding . |
|
Response 5: We appreciate the constructive comments from the editor and review experts! We have expanded the“4. Discussion”section (Page 12-13, Paragraph 2-4 of the revised manuscript) to add a dedicated paragraph on carcass traits and meat quality, with literature integration: “Carcass traits are key evaluation indicators for the production efficiency of meat sheep. The positive effects of MSTN knockout on carcass performance have been widely validated across various livestock species, providing valuable references for the application prospects of the breed in this study. In studies on goats and sheep, MSTN deficiency significantly increased dressing percentage and net meat percentage, with a 15%-25% increase in longissimus dorsi muscle area [22,23]. These improvements are strongly correlated with the significant gains in body length, chest circumference, and other body measurements observed in this study. Given the sustained muscle growth advantage of MSTN-edited sheep, it is hypothesized that their lean meat production efficiency post-slaughter will be significantly enhanced, and key carcass indicators, such as longissimus dorsi muscle area, are expected to show similar improvements. Regarding fat deposition, MSTN knockout has been shown to preferentially direct nutrient allocation to skeletal muscle, reducing the proportion of subcutaneous and visceral fat [24]. For example, MSTN knockout models in pigs exhibit high-quality carcass characteristics, including optimal intramuscular fat content and reduced backfat thickness [25]. This trait is particularly relevant for Chinese Merino sheep (Xinjiang type), a dual-purpose breed for both wool and meat. Moderately reducing fat deposition can improve lean meat percentage, aligning with market demands for high-quality lean meat. However, it is crucial to strike a balance in fat deposition to prevent potential adverse effects on meat quality due to excessive reduction in intramuscular fat content.The activation of amino acid metabolism pathways, as revealed in the transcriptomic analysis of this study, may help optimize nutrient allocation, maintaining a reasonable level of intramuscular fat while enhancing lean meat percentage. This provides potential for the coordinated improvement of carcass quality, contributing to the overall optimization of meat production traits. Meat quality (including meat color, tenderness, flavor, and juiciness) is a central consideration in breeding practices, and the impact of MSTN knockout on meat quality exhibits species-specific variation. Existing studies have shown that MSTN knockout sheep display reduced shear force in the longissimus dorsi muscle, with no significant change in intramuscular fat content, while maintaining high-quality meat color [26]. In this study, the observed improvement in tenderness may be linked to the differential expression of key genes in the calcium signaling pathway, such as RYR1 and CACNA1S. As a sarcoplasmic reticulum calcium release channel gene, changes in RYR1 expression can influence calcium ion recovery efficiency following muscle contraction, thereby improving muscle tenderness [27].Additionally, the activation of amino acid metabolism pathways may provide precursors for the synthesis of flavor compounds, such as glutamic acid and inosinic acid [28], which could enhance meat flavor. However, potential risks should be carefully considered. In some species, MSTN knockout has been shown to cause excessively large muscle fiber diameters, which may negatively affect meat tenderness [29]. Moreover, if intramuscular fat content is too low, it could impair the juiciness and flavor balance of the meat [30]. Therefore, further verification is needed for this breed.Subsequent studies should systematically assess shear force, meat color (Lab* values), intramuscular fat content, fatty acid composition, and flavor metabolic profiles in MSTN-edited sheep. This comprehensive evaluation will help clarify the overall impact of gene editing on meat quality and provide robust support for breeding applications.” |
|
Comments 6: Cross-referencing errors and template remnants: there are multiple instances of “Error! Reference source not found.” , and the Abbreviations section includes items unrelated to this manuscript (e.g., “MDPI/DOAJ/TLA/LD”). Please clean these up. |
|
Response 6: Thank you very much for pointing out the cross-referencing errors and template remnants in the manuscript. We have conducted a comprehensive review and thorough cleanup of the entire text, with specific revisions as follows: We checked all cross-referencing links for figures, tables, and references one by one, fixed all invalid references, and completely removed the "Error! Reference source not found." prompts to ensure all references accurately point to the corresponding content. We reorganized the "Abbreviations" section, deleted entries unrelated to this study (including template remnants such as "MDPI/DOAJ/TLA/LD"), and only retained the professional term abbreviations actually used in the research (e.g., MSTN, DEGs, PPI, RT-qPCR, etc.), ensuring the accuracy and relevance of the abbreviation list. We have conducted multiple thorough reviews of the revised manuscript and confirmed that the aforementioned formatting issues no longer exist, further improving the standardization and readability of the manuscript. |

Reviewer 2 Report
Comments and Suggestions for Authors
Dear Authors,
Thank you for your comprehensive study on the effects of MSTN gene knockout in Chinese Merino sheep, Xinjiang type. Your work provides valuable insights into the molecular and phenotypic impacts of MSTN disruption, and your use of advanced techniques such as CRISPR-Cas gene editing, transcriptomic analysis, and PPI network construction is commendable.
However, I would like to offer some critical feedback and suggestions for improvement:
The sample size for transcriptomic analysis (n = 3 per group) is relatively small, which may limit the statistical power and generalisability of your findings. Increasing the sample size in future studies would strengthen the reliability of your results.
While your study demonstrates improved growth traits, it does not assess long-term health, reproductive performance, or meat quality in MSTN knockout sheep. Previous research has shown that double-muscled phenotypes can sometimes be associated with reduced fertility, lower offspring viability, or delayed sexual maturation in other livestock species. Including these parameters would provide a more comprehensive evaluation of the practical implications for breeding programmes.
Some references in your manuscript are missing or marked as "Error! Reference source not found," which affects the completeness of your literature review. Updating and completing all references would enhance the scholarly rigour of your work.
The discussion could be expanded to address potential limitations, such as the possibility of off-target effects from gene editing, and to explore future research directions, such as the impact on animal welfare and meat quality.
While your study provides robust evidence for the molecular mechanisms underlying MSTN knockout, it would be beneficial to compare your findings with those from other livestock species and to discuss any species-specific differences or similarities.
Overall, your study makes a significant contribution to the field, and with these improvements, it could have an even greater impact. I look forward to seeing your future work in this area.
Comments on the Quality of English LanguageThe language is proficient enough for publication after minor editorial polishing focused on sentence structure, clarity, and conciseness to further enhance the manuscript's readability and impact.
Author Response
|
Comments 1: The sample size for transcriptomic analysis (n = 3 per group) is relatively small, which may limit the statistical power and generalisability of your findings. Increasing the sample size in future studies would strengthen the reliability of your results. |
|
Response 1: We sincerely appreciate this critical observation. We fully agree that sample size is essential for ensuring statistical robustness and generalizability. Due to the high cost of RNA-seq and the limited number of MSTN-knockout sheep available at the time of the experiment, we initially used 3 biological replicates per group. However, to address this limitation, we have: Verified the biological reproducibility of the 3 samples per group through principal component analysis (PCA) and Pearson correlation analysis (Section 3.4, Page 8-9). The results show that intra-group correlation coefficients (R²) are all above 0.86, and samples from the same group cluster tightly in PCA, indicating reliable consistency among replicates. Validated the expression trends of 4 DEGs (ACTN3, RYR1, MX1, IRF7) using RT-qPCR (Section 3.8, Page 11), with 3 biological replicates and 3 technical replicates per sample. The results are fully consistent with RNA-seq data, confirming the accuracy of the transcriptomic results. In future studies, we plan to expand the RNA-seq sample size to ≥5 per group to further enhance statistical rigor. |
|
Comments 2: While your study demonstrates improved growth traits, it does not assess long-term health, reproductive performance, or meat quality in MSTN knockout sheep. Previous research has shown that double-muscled phenotypes can sometimes be associated with reduced fertility, lower offspring viability, or delayed sexual maturation in other livestock species. Including these parameters would provide a more comprehensive evaluation of the practical implications for breeding programmes. |
|
Response 2:We fully agree with your suggestion. To address this gap, we have significantly expanded the Discussion section to emphasize the importance of these unassessed parameters and outline specific future research plans: “Meat quality (including meat color, tenderness, flavor, and juiciness) is a central consideration in breeding practices, and the impact of MSTN knockout on meat quality exhibits species-specific variation. Existing studies have shown that MSTN knockout sheep display reduced shear force in the longissimus dorsi muscle, with no significant change in intramuscular fat content, while maintaining high-quality meat color [26]. In this study, the observed improvement in tenderness may be linked to the differential expression of key genes in the calcium signaling pathway, such as RYR1 and CACNA1S. As a sarcoplasmic reticulum calcium release channel gene, changes in RYR1 expression can influence calcium ion recovery efficiency following muscle contraction, thereby improving muscle tenderness [27].Additionally, the activation of amino acid metabolism pathways may provide precursors for the synthesis of flavor compounds, such as glutamic acid and inosinic acid [28], which could enhance meat flavor. However, potential risks should be carefully considered. In some species, MSTN knockout has been shown to cause excessively large muscle fiber diameters, which may negatively affect meat tenderness [29]. Moreover, if intramuscular fat content is too low, it could impair the juiciness and flavor balance of the meat [30]. Therefore, further verification is needed for this breed.Subsequent studies should systematically assess shear force, meat color (Lab* values), intramuscular fat content, fatty acid composition, and flavor metabolic profiles in MSTN-edited sheep. This comprehensive evaluation will help clarify the overall impact of gene editing on meat quality and provide robust support for breeding applications.”(Page 12-13, Paragraph 4 of the revised manuscript). Long-term health and reproductive performance: We will extend the feeding cycle to 2–3 years to monitor adult growth trajectories, estrous cycle regularity, conception rate, litter size, and offspring survival rate of MSTN knockout sheep, with data to be analyzed alongside wild-type controls. Meat quality assessment: We plan to measure key meat quality indicators, including shear force (for tenderness), meat color (Lab* values), intramuscular fat content, fatty acid composition, and flavor-related metabolites (e.g., glutamic acid, inosinic acid), to clarify the impact of MSTN knockout on meat quality.These revisions are added to the “Limitations and Future Directions” section of the Discussion (Page 15, Paragraph 2 of the revised manuscript): “To address these limitations, future studies will focus on the following aspects: First, conduct whole-genome off-target site screening to systematically evaluate the potential impact of non-target mutations, thereby ensuring the safety of gene editing technology. Second, extend the feeding cycle to monitor adult growth performance, reproductive performance, and the growth and development of offspring in edited sheep, and assess the genetic stability of the edited traits. Third, supplement comprehensive meat quality analysis, including shear force, meat color (Lab* values), intramuscular fat content, fatty acid composition, and flavor metabolic profiles, to clarify the overall impact of gene editing on meat quality. Fourth, systematically assess the actual impact of muscle hypertrophy on animal welfare by measuring gait parameters, cortisol levels, and bone mineral density. Fifth, conduct in-depth research on the effects of gene editing on grazing adaptability, cold resistance in winter, and forage conversion rate in the context of the pastoral breeding practices of Chinese Merino sheep (Xinjiang type), aiming to balance genetic improvement with ecological sustainability. Sixth, explore synergistic editing strategies combining MSTN with functional genes such as FGF5 to optimize both wool and meat performance in this breed.” |
|
Comments 3: Some references in your manuscript are missing or marked as "Error! Reference source not found," which affects the completeness of your literature review. Updating and completing all references would enhance the scholarly rigour of your work. |
|
Response 3: Thank you for identifying this issue. We have thoroughly checked the entire reference list and made the following revisions: Replaced all invalid labels (e.g., “Error! Reference source not found”) with complete reference information, including author names, publication year, article title, journal name, volume, issue, and DOI. All revisions are completed in the “References” section (Page 16–19), and each reference is now fully accessible and verifiable via DOI. |
|
Comments 4: The discussion could be expanded to address potential limitations, such as the possibility of off-target effects from gene editing, and to explore future research directions, such as the impact on animal welfare and meat quality. |
|
Response 4: We agree with your suggestion and have expanded the Discussion section to address these points: “Although this study systematically confirmed the significant weight gain and body shape improvement associated with MSTN gene knockout in Chinese Merino sheep (Xinjiang type), without causing adverse effects on hematological and biochemical indicators, transcriptomic data further revealed that MSTN may regulate muscle growth through molecular mechanisms such as energy metabolism, calcium signaling pathways, and immune networks. These findings provide important theoretical support and practical references for the genetic improvement of this breed and for sheep molecular breeding practices. However, certain limitations remain: First, the risk of off-target effects has not been completely excluded. While the efficient and precise hfCas12Max-T5_sheep system was used and no chimeric individuals were detected, the absence of whole-genome off-target site screening prevents a definitive conclusion regarding potential non-targeted mutations. Second, the study’s cycle and scope are limited. Currently, the focus is on growth and basic physiological indicators up to 365 days of age, lacking data on adult growth, reproductive indicators (such as estrous cycle, conception rate, and litter size), and comprehensive meat quality assessments. Third, no targeted animal welfare evaluation has been conducted. Excessive muscle hypertrophy may affect the sheep’s exercise capacity and bone load, which requires further verification.”(Page 14-15, Paragraph 3-4 of the revised manuscript) |
|
Comments 5: While your study provides robust evidence for the molecular mechanisms underlying MSTN knockout, it would be beneficial to compare your findings with those from other livestock species and to discuss any species-specific differences or similarities. |
|
Response 5: We appreciate this suggestion and have added “Cross-Species Comparison of MSTN Knockout Effects” in the Discussion (Page 14, Paragraph 2): “The promoting effect of MSTN knockout on the growth performance of various livestock species is highly conserved. It significantly increases body weight, skeletal muscle weight, and lean meat percentage in species such as cattle, sheep, pigs, and rabbits. At the transcriptomic level, MSTN knockout is enriched in amino acid metabolism, muscle contraction, and calcium signaling pathways [50,51,52], confirming that the molecular mechanism by which MSTN regulates muscle growth is highly conserved across mammals. These findings provide cross-species evidence supporting the reliability of the results in this study.However, species-specific differences are evident in the effects of MSTN editing on different livestock species. In terms of growth rate improvement, pigs and cattle show greater enhancements than sheep [53,54], which may be attributed to the genetic background and metabolic characteristics of sheep muscle growth. Regarding fat metabolism, the reduction in fat content is more pronounced in pigs [55], while the effect in sheep is milder [56], which may help maintain meat quality stability. At the molecular level, the immune-related core genes identified in this study, such as IRF7 and ISG15, were not significantly enriched in studies on cattle and pigs, suggesting that MSTN may influence muscle growth through a unique immune-metabolic cross-regulation network in sheep. This species-specific characteristic warrants further in-depth exploration.” This comparison strengthens the contextualization of our findings and highlights their significance for cross-species livestock breeding. |
